

# Laboratory and clinical trials of cocamide diethanolamine lotion against head lice

Ian F. Burgess, Elizabeth R. Brunton and Christine M. Brown

Medical Entomology Centre, Insect Research & Development Limited, Cambridge, United Kingdom

## ABSTRACT

**Context.** During the late 1990s, insecticide resistance had rendered a number of treatment products ineffective; some companies saw this as an opportunity to develop alternative types of treatment. We investigated the possibility that a surfactant-based lotion containing 10% cocamide diethanolamine (cocamide DEA) was effective to eliminate head louse infestation.

**Settings and Design.** Initial *in vitro* testing of the lotion formulation versus laboratory reared body/clothing lice, followed by two randomised, controlled, community-based, assessor blinded, clinical studies.

**Materials and Methods.** Preliminary laboratory tests were performed by exposing lice or louse eggs to the product using a method that mimicked the intended use. Clinical Study 1: Children and adults with confirmed head louse infestation were treated by investigators using a single application of aqueous 10% cocamide DEA lotion applied for 60 min followed by shampooing or a single 1% permethrin creme rinse treatment applied to pre-washed hair for 10 min. Clinical Study 2: Compared two treatment regimens using 10% cocamide DEA lotion that was concentrated by hair drying. A single application left on for 8 h/overnight was compared with two applications 7 days apart of 2 h duration, followed by a shampoo wash.

**Results.** The initial laboratory tests showed a pediculicidal effect for a 60 min application but limited ovicidal effect. A longer application time of 8 h or overnight was found capable of killing all eggs but this differed between batches of test material. Clinical Study 1: Both treatments performed badly with only 3/23 (13%) successful treatments using cocamide DEA and 5/25 (23.8%) using permethrin. Clinical Study 2: The single overnight application of cocamide DEA concentrated by hair drying gave 10/56 (17.9%) successes compared with 19/56 (33.9%) for the 2 h application regimen repeated after 1 week. Intention to treat analysis showed no significant difference ($p = 0.0523$) between the treatments. Over the two studies, there were 18 adverse events possibly or probably associated with treatment, most of which were increased pruritus after treatment.

**Conclusions.** Cocamide DEA 10% lotion, even when concentrated by hair drying, showed limited activity to eliminate head louse infestation.

Corresponding author
Ian F. Burgess,
ian@insectresearch.com

## INTRODUCTION

Interest in the use of plant derived pediculicides, mostly essential oils, was rekindled following the discovery of head lice resistant to insecticides in the 1990s (*Veal, 1996*; *Yang et al., 2004*; *Heukelbach, Speare & Canyon, 2009*). These developments also included non-volatile fixed vegetable oils, with anecdotal claims of effectiveness for olive oil, mayonnaise, margarine, and coconut oil, which are messy to use and have doubtful effectiveness. Unmodified plant oils have been used for centuries as hair conditioners in southern Europe, South Asia, and Africa, without affecting lice as confirmed by laboratory tests of so-called "home remedies" (*Takano-Lee et al., 2004*). However, modified vegetable oil surfactants are widely used in toiletry cleansing products to remove oils and other materials from hair. Some of these may dissolve waterproofing lipids that protect the louse cuticle from dehydration but there has been little interest in evaluating them clinically for effectiveness.

Cocamide diethanolamine (cocamide DEA) is a surfactant that stabilises the foam in "stripping" or "clarifying" shampoos and is highly efficient at removing lipids and other deposits from the hair. It has been used in head louse treatments, mainly as an excipient but also an active ingredient, commonly described as modified coconut oil. During the early 1990s several "modified coconut oil" products were sold in Central Europe, but efficacy studies were poorly reported with unclear methodologies (*Mülhofer, 1994*). One manufacturer suggested the material asphyxiated lice by blocking the spiracles; assuming that coconut oil has an occlusive effect. In reality, this potent surfactant is more likely to disrupt the cuticular lipid of the lice.

We have conducted an investigation of the activity of cocamide DEA in the treatment of head louse infestation. Prior to initiating clinical studies, the material was tested *in vitro* to confirm that the sponsor's formulation was active and to establish whether the approach to dosing, as set out in a previous report (*Mülhofer, 1994*), was appropriate. The initial randomised study was planned to show equivalence of the cocamide DEA 10% lotion with 1% permethrin creme rinse. However, this study was not successful so further *in vitro* tests were performed to see if a more effective treatment regimen was possible using this formulation. As a result of these tests, two possible treatment options for the cocamide DEA preparation were selected and subsequently tested in a second clinical study.

## METHODS AND MATERIALS

### Pre-clinical studies

We performed pre-clinical laboratory evaluations of cocamide DEA in a manner intended to mimic as closely as possible use of the product by a consumer in essentially the same way as described previously for other products (*Burgess, 1991*; *Burgess et al., 2012*). We used laboratory reared human clothing/body lice, *Pediculus humanus humanus*, to conduct the *in vitro* tests against adult insects. The lice were given squares of nylon gauze substrate throughout the tests. We provided adult lice with nylon gauze as a substrate for oviposition so we could handle the eggs without risk of damaging them and the effects of the treatment could be observed easily.

Before the first clinical study we performed tests to check the claims made previously by *Mülhofer (1994)* that the product was completely effective when applied for 60 min. In these tests we compared two potential formulations of aqueous cocamide DEA with 3.5% and 10% active substance, using three batches of 20 lice on 15 × 15 mm squares of gauze as the test insects. In this test method, we fully immersed the insects in the product for 10 s then, using forceps, lifted the gauze bearing the insects from the fluid, which allowed excess liquid to drain off as it would if the lotion had been poured onto the scalp. We then incubated them in closed 55 mm diameter plastic Petri dishes for 60 min, still wetted with the test fluid, followed by washing with non-medicated shampoo (Boots frequent wash shampoo), followed by rinsing and blotting dry. The lice were maintained overnight in a humidified incubator (30 °C ± 2 °C and 50% relative humidity) and the effects of the treatment were recorded 24 h after the initial exposure. For this first investigation freshly laid eggs were treated in the same way as lice, but in this case using only the 10% cocamide DEA mixture and then, after washing and rinsing, incubated at 30 °C ± 2 °C and 50% humidity until the untreated control group of eggs had completed hatching, approximately 10 days later. In both louse and egg tests, the control groups were treated to the same procedures except we used tap water in place of the test formulation.

The evaluation of the effect of treatments on lice determined whether they were alive and moving normally ("Mobile"); not walking but possibly exhibiting small movements of appendages or peristalsis of the gut ("Immobile"); or immobile with no detectable sign of residual life ("Killed"). Both "Immobile" and "Killed" were included in mortality figures. For louse eggs, those that had developed fully and from which a living louse nymph had emerged completely were recorded as "Hatched." In some cases the nymph had developed and started the emergence process but had died before escaping from the eggshell ("Half-hatched"). Eggs in which there was no obvious sign of embryonic development, specifically no appearance of the eyespot, which would normally become apparent between the third and fourth day of incubation, were designated "Undeveloped," whereas eggs that contained an embryo with eyespots but that failed to hatch were recorded as having "Died." In the case of louse eggs, both "Died" and "Undeveloped" were included in mortality figures.

In preparation for the second clinical study we ran a series of tests against louse eggs, using essentially the same methods, to compare different treatment regimens of exposure time (1 h, 2 h, or overnight); washing with shampoo followed by rinsing or washing with water only; and the influence of hair drying over a period of approximately 5 min, using an electric hair dryer on "Cool" setting held at a variable distance of approximately 30 cm from the test specimens, compared with humidification after application of the product. The temperature during hair drying was modulated by placing the fingers of the operator's hand between the air jet and the test insects, and by movement of the device nearer or further away during the drying process so that a temperature of no greater than 40° C occurred in the dish where the insects were held, measured using a thermocouple probe. Also, we compared the sensitivity of louse eggs of different ages to the product by collecting eggs on a single day, storing them under the same conditions and then treating randomly

sampled gauze squares, each bearing approximately 100 eggs, on the subsequent test days. As with the first series of laboratory tests, the control groups were treated in the same way but using tap water for the initial exposure to fluid.

After completion of the second clinical study, the low level of efficacy achieved, despite using longer application times and increasing the dose concentration of the cocamide DEA by evaporation, caused us to consider the possibility that the two batches of cocamide DEA lotion were different in some way. We were able to do this by performing further *in vitro* tests in which different groups of louse eggs from the same batch were treated with samples the two batches of lotion, using both 2 h and overnight applications followed by a water rinse.

## Clinical studies
### Study medications
We conducted two randomised, controlled, assessor blinded clinical studies, using essentially the same procedures throughout, although the treatments offered were different. In both studies the investigative product was a preserved aqueous solution of cocamide diethanolamine (cocamide DEA) 10%. This preparation was supplied in 100 mL polyethylene bottles with a pour-on dispenser, applied systematically to saturate the hair and scalp, and massaged in. Investigators applied all the treatments.

In the first study, one group of participants was randomised to receive cocamide DEA 10% (batch number LI 35101) applied for 60 min followed by washing with non-medicated shampoo (Boots frequent wash shampoo) then rinsing with water. The comparison group was treated using 1% permethrin creme rinse (Lyclear creme rinse, Chefaro UK Ltd, Huntingdon, UK) applied for 10 min to pre-washed and towel dried hair, followed by rinsing. Both products were applied on a single occasion. Shampoo was supplied to both groups by the investigators applying the treatment. No guidance on routine hair washing was given for this study.

The second study compared two treatment regimens using cocamide DEA 10% (batch number NH 35101). In one group we applied the product to dry hair until it was thoroughly soaked, then evaporated the excess water using a hair dryer leaving the hair sticky, with the appearance of having been heavily oiled. The length of the "drying" process varied according to the length and thickness of the hair and ranged from approximately 3–20 min, which was determined by the appearance and feel of the treated hair. This was then left overnight then washed off by the participant or parent with plain warm water in the morning. There was no second treatment.

For the other group, we also applied the product to dry hair and dried off excess water with the hair dryer to the same sticky stage. This was then left for a timed 2 h period before being rinsed off with plain warm water by the investigator. A repeat treatment was given 7 days later.

For both groups, participants were given bottles of the same non-medicated shampoo to use for normal hair washing on the third and tenth days after treatment.

**Table 1 Activity of two concentrations of cocamide DEA lotion against lice *in vitro*.**

| Treatment | Number of lice from three replicates of 20 | | | | |
|-----------|-------|--------|----------|--------|-------------|
|           | Total | Killed | Immobile | Mobile | Mortality % |
| Cocamide DEA 3.5% | 62 | 51 | 6 | 5 | 91.9 |
| Cocamide DEA 10% | 62 | 54 | 7 | 1 | 98.4 |
| Water control | 62 | 5 | 4 | 53 | 14.5 |

**Table 2 Activity of 10% cocamide DEA lotion against freshly laid louse eggs *in vitro*.**

| Treatment | Number of eggs from three replicate tests | | | | | Mortality % |
|-----------|-------|---------|-------------|------|-------------|-------------|
|           | Total | Hatched | Half-hatched | Died | Undeveloped | |
| Cocamide DEA 10% | 569 | 3 | 1 | 495 | 70 | 99.3 |
| Water control | 518 | 458 | 3 | 19 | 38 | 11.0 |

## *Participants*

Participants were recruited from respondents to an invitation letter and information sheet distributed through schools or via general practitioners. Parents of children with lice telephoned the study co-ordinator and made an appointment for a home visit. We visited within 24 h to check prospective participants for living lice using a plastic detection comb (Albyn of Stonehaven, Stonehaven, Scotland). If an infestation was found, consisting of one or more live lice, the person was invited to join the study and the parent or guardian guided through a written consent procedure followed by child assent. All household members were offered examination and, if found to be infested, the opportunity to join the study if fitting the enrolment criteria. In this context, the intensity of an infestation was partially subjective: Heavy = more than one louse found with the first stroke of the comb; Medium = one louse found with the first stroke; and Light = lice found only after several strokes of the comb over different parts of the head. Lice were returned to the head because treatment followed. The number of participants with each level of infestation is shown in Tables 1 and 2.

Consenting participants provided baseline demographic data including age, gender, hair characteristics, concurrent medications, and medical history. Some demographic characteristics, such as hair dryness and thickness, were subjective assessments made by the investigators on the day and were intended to serve only as a guideline. In both studies, the lower age limit was 4 years. People who were sensitive to paraben preservatives; who had persistent disorders of the scalp such as eczema or psoriasis; who had received a head louse treatment within the 4 weeks prior to entry; or had undergone antibiotic treatment or had their hair bleached, colour treated, or permanently waved within the previous 4 weeks, were excluded. All treatments and assessments were performed in the participant's home. No payment was offered for participation.

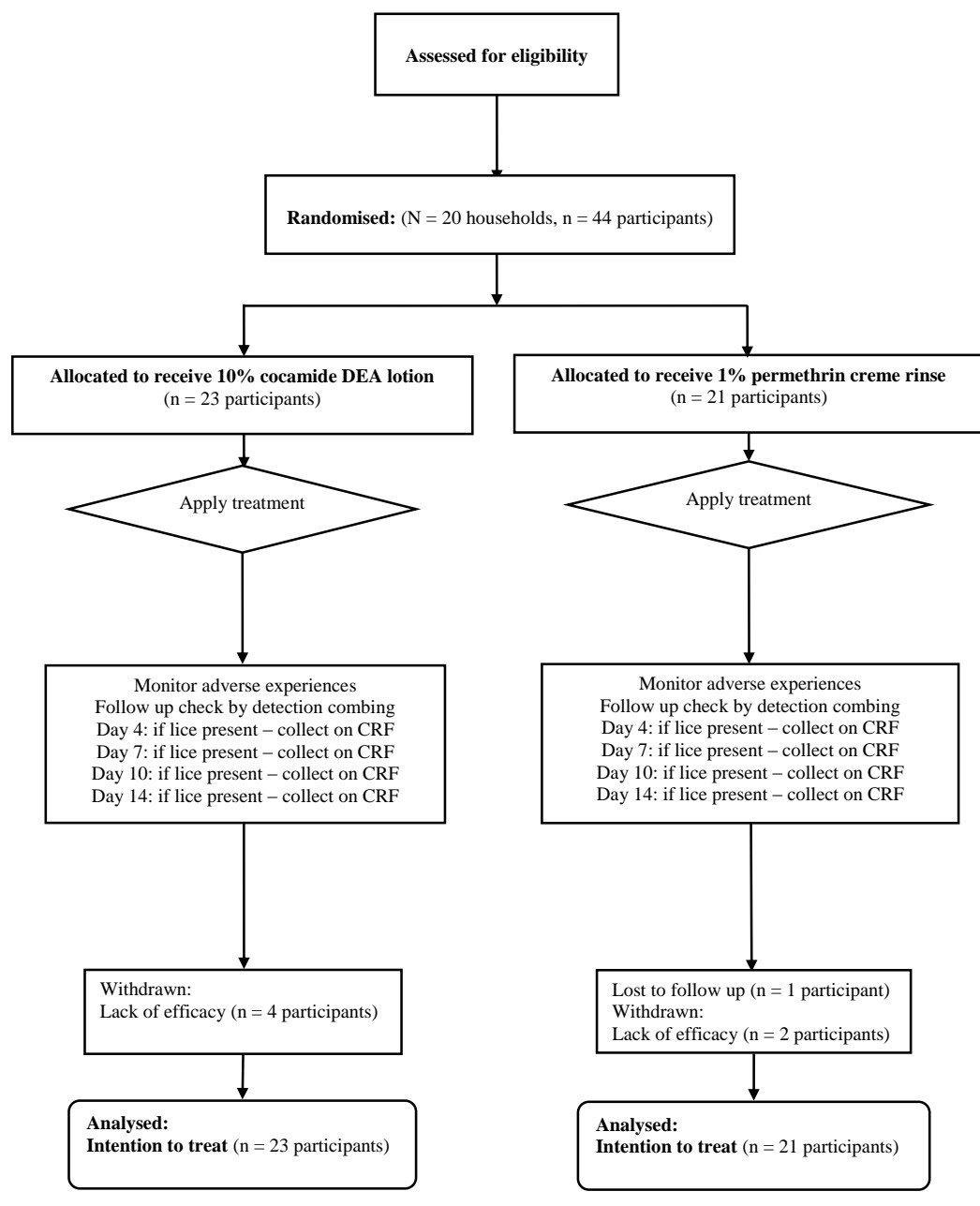

**Figure 1** **Flowchart of participants through the first clinical study.**

For the first study, between July 1998 and January 1999, we recruited 44 participants (35 children and 9 adults), out of the 120 planned, divided equally between cocamide DEA 10% and 1% permethrin creme rinse treatments (Fig. 1). Of these, one person treated with permethrin was lost to follow up before any post-treatment checks could be made. Six others (four treated with cocamide DEA and two with permethrin) were withdrawn at the participant's request due to lack of efficacy. This left 19 participants treated with cocamide

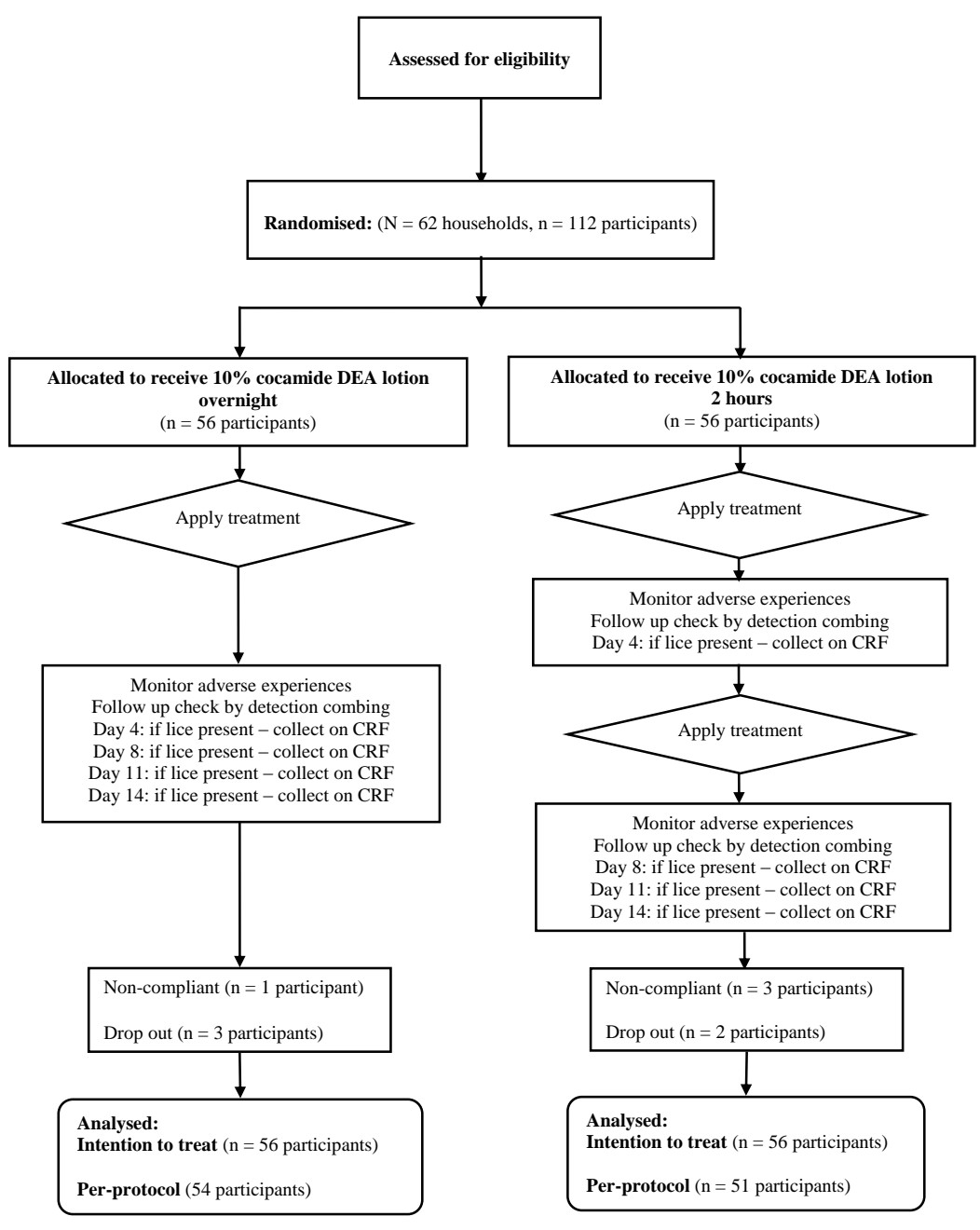

**Figure 2 Flowchart of participants through the second clinical study.**

DEA and 18 with permethrin with complete data sets before the sponsor requested an early termination on grounds of lack of efficacy for both products.

In the second study, we recruited 112 out of 120 planned enrolments between October and December 1999. All participants were treated with cocamide DEA 10% divided between two treatment regimens: either a single overnight application or two 2 h applications a week apart. There were seven participants withdrawn (Fig. 2). Three people, one from the single application group and two from the double application

group, dropped out because a baby from the family was hospitalised and the others were unable to keep appointments. Four participants were withdrawn by investigators due non-compliance, one from the single application group and three from the double application group. This left 54 people treated with one overnight application and 51 who received two 2 h applications. Recruitment for this study was also terminated early at the sponsor's request on grounds of lack of efficacy.

### Ethics

Ethical approval was granted by the Huntingdon Local Research Ethics Committee of Cambridgeshire Health Authority for the first study (protocol CTRL01, REC reference 97/229, dated 25th July 1997). The second study received approval from both Huntingdon LREC and Cambridge Local Research Ethics Committee (protocol CTRL02, REC reference 99/211, dated 17th September 1999). A Clinical Trial Exemption Certificate (CTX16442/0001/A) was granted by the UK Medicines Control Agency to permit conduct of each of the trials.

In each study, parents/guardians stated that they understood the purpose of the study and they agreed to abide by the requirements of the protocol before providing witnessed written consent for participating children under the age of 16 years.

The studies were conducted in conformity with the principles of the Declaration of Helsinki and the ICH Guidelines for Good Clinical Practice (GCP) prevailing at the time.

These studies were not registered on a publically accessible database at the time they were conducted because there was no facility at the time for doing so. Both studies have now been retrospectively registered on the database at https://clinicaltrials.gov with the registration number NCT02500524 for the cocamide DEA versus permethrin study (CTRL01) and NCT02499549 for the comparison between treatment regimens using cocamide DEA (CTRL02).

### Outcome measures

For both studies the primary outcome measure was elimination of infestation following completion of the allocated treatment regimen. We used a plastic louse detection comb on dry hair for follow up examinations looking for lice. In the first study, assessments were made on Days 4, 7, 10 and 14 after treatment. A similar approach was taken for the second study but, because one arm of the study used a two application regimen, the timing was adjusted so that checkups were made on Days 4, 8, 11, and 14 after the first application of treatment. Any live lice found during these visits were collected on the case record forms.

### Sample size

Both studies were designed to demonstrate equivalence to within 20% between treatments. Sample sizes were estimated on the basis that it would be possible to detect equivalence with 95% confidence, assuming that the underlying rates of efficacy would be 90%. At this level, we estimated that a group size of 50 would provide 80% power and 61 would give 90% power. Therefore, a sample size of 60 per group was selected on the basis that it would provide at least 85% power, even allowing for possible post-randomisation protocol violations. The estimate also showed that if the underlying rates of the treatments averaged

90%, this sample size would also have a power of 90% to detect a difference of around 18% with 95% confidence.

### Randomisation and allocation concealment

Randomisation codes for both studies were produced from computer generated lists by an independent statistician appointed by the sponsor. Allocation sequences were made in balanced blocks of 12 and the identification of each treatment entered on instruction sheets enclosed in sealed, numbered, opaque envelopes. The study numbers were allocated in sequence, and the identity of the allocated treatment only revealed after receipt of informed consent and confirmation of suitability to participate in the study. Post-treatment assessments were blinded and performed by different investigators unaware of the allocated treatment.

### Statistical methods

The outcomes of tests conducted *in vitro* were analysed using a purpose built calculator for estimating the 95% confidence interval (CI) from the normal approximation to the binomial distribution. Overall, with the high level of treatment failure in each of the treatments arms in both clinical studies, combined with the sponsor request to terminate both studies early, the intended analyses could not be performed in the way intended. However, we were able to analyse differences between the groups in terms of efficacy, safety, ease of use, and acceptability using the intention to treat population for both studies by Fisher's exact test and unstratified chi-squared tests for yes/no variables and the Kruskal-Wallis test for ranked variables. The initial analyses were expected to be performed by the consultant statistician (PN Lee Statistics and Computing Ltd, Sutton, UK) using bespoke software, but because the studies were terminated early the sponsor decided not to proceed with this work. We also planned to test for equivalence using the per-protocol population based on 95% confidence limits derived from the normal approximation. However, because both studies were terminated early a true per-protocol group was not determined.

## RESULTS

### Pre-clinical studies

In the first series of *in vitro* tests, comparing the 3.5% and 10% lotions that we performed prior to clinical studies, we found the 10% mixture was the more effective using an exposure for 60 min followed by washing with shampoo that resulted in the death of all but one laboratory reared clothing/body lice when reviewed the following day (Table 1). The effect on louse eggs was also encouraging indicating that a high proportion of eggs could be inhibited from hatching, with a significant ($p < 0.01$, 95% CI [0.014–0.085]) increase in the proportion of eggs that showed no sign of development post-treatment compared with the control group, suggesting penetration of the egg structure by the active component of the lotion (Table 2).

Since it was clear from the first clinical study that inability to kill the louse eggs contributed significantly to the failure to cure, we conducted a second *in vitro* study to

**Table 3 Effect of different application times using 10% cocamide DEA lotion on louse eggs at different development ages.**

| Age of eggs— application time | Number of eggs from three replicate tests | | | | | Mortality % (Undeveloped %) |
|---|---|---|---|---|---|---|
| | Total | Hatched | Half-hatched | Died | Undeveloped | |
| **Cocamide DEA** | | | | | | |
| 1 day old—2 h | 353 | 0 | 0 | 56 | 297 | 100 (84.1) |
| 1 day old—overnight | 291 | 0 | 0 | 3 | 288 | 100 (99.0) |
| 4 days old—2 h | 278 | 15 | 20 | 189 | 54 | 94.6 (19.4) |
| 4 days old—overnight | 283 | 0 | 13 | 43 | 227 | 100 (80.2) |
| 5 days old—2 h | 278 | 0 | 38 | 71 | 169 | 100 (60.8) |
| 5 days old—overnight | 300 | 0 | 0 | 218 | 82 | 100 (30.2) |
| 6 days old—2 h | 271 | 84 | 90 | 82 | 14 | 69.0 (5.2) |
| 6 days old—overnight | 324 | 0 | 1 | 293 | 30 | 100 (9.3) |
| 7 days old—overnight | 231 | 47 | 42 | 121 | 21 | 79.7 (9.1) |
| 8 days old—overnight | 137 | 28 | 2 | 101 | 6 | 79.6 (4.4) |
| Water control | 328 | 310 | 4 | 4 | 10 | 5.5 (3.1) |

investigate whether older eggs were harder to kill. In the original *in vitro* test louse eggs 24–48 h old were used, whereas on the head there would be eggs at different development stages from newly laid through to hatching. We found that, as the eggs aged, they became less susceptible to the treatment, with 5 day and older eggs requiring exposures longer than 2 h to stop louse nymphs from emerging and by the seventh day not all eggs were prevented from hatching using an overnight exposure. In addition, the proportion of embryos that failed to develop eyespots (the first definitive indication of embryonic development) was lower when exposed to a 2 h treatment than overnight and by the 6th day there was no difference from the control group because all embryos that would develop had reached the "eyespot" stage between days 5 and 6 (Table 3).

Increasing the dose concentration of cocamide DEA, by prolonging the treatment time or speeding up the evaporation rate of the excipient water using a hair dryer, produced some increase in activity, with no significant increase in overall inhibition of hatching but with a significant ($p < 0.001$, 95% CI [0.131–0.266]) increase in the proportion of eggs failing to develop after hair drying, whereas inhibiting water evaporation in a saturated atmosphere appeared to reduce the activity (Table 4). Use of shampoo to remove the product rather than simple water rinsing was potentially the most important factor for reduction of activity ($p = 0.0004$, 95% CI [0.078–0.304]), especially against older louse eggs, which showed no significant difference from the control group on overall failure to hatch (Table 4).

## Clinical studies
### Participant flow
Over the length of the two studies, we recruited participants from 82 households ranging in size from 2 to 8 members (mean 4.56), with between 1 and 5 people (mean 1.93) enrolled.

**Table 4 Effect of different washing or incubation regimens on effect of cocamide DEA lotion on louse eggs.**

| Treatments | Number of eggs from three replicate tests | | | | | Mortality % (Undeveloped %) |
|---|---|---|---|---|---|---|
| | Total | Hatched | Half-hatched | Died | Undeveloped | |
| **Cocamide DEA** | | | | | | |
| 1 day old eggs | | | | | | |
| 60 min—shampoo wash | 117 | 80 | 1 | 30 | 6 | 31.6 (5.1) |
| 60 min—water rinse | 205 | 101 | 5 | 92 | 7 | 50.7 (3.4) |
| 120 min—water rinse | 111 | 7 | 1 | 94 | 9 | 93.7 (8.1) |
| 60 min—humidified + water rinse | 247 | 139 | 6 | 95 | 7 | 43.7 (2.8) |
| 60 min + drying—water rinse | 159 | 64 | 0 | 58 | 37 | 59.8 (23.3) |
| 5 day old eggs | | | | | | |
| 60 min—water rinse | 117 | 100 | 7 | 3 | 7 | 14.5 (6.0) |
| Water control | 203 | 187 | 2 | 6 | 8 | 7.9 (3.9) |

In both studies, the majority of households had either four or five members. In the first study, comparing cocamide DEA with permethrin, there were no significant demographic differences between the groups (Table 5). A similar demographic was observed in the second study, comparing the two treatment regimens of cocamide DEA, with the exception of a significant ($p < 0.038$) difference in proportion of 10–14 year old participants (Table 6) and a non-significant trend ($p = 0.073$) in the single application group for more people to have "Average" thickness hair.

### Outcomes

In the comparison between a single 1 h application of cocamide DEA 10% lotion and one 10 min application of 1% permethrin creme rinse, we found no significant difference between the treatments. Both treatments performed badly in terms of efficacy with success in 3/23 (13.0%) of participants treated with cocamide DEA (1 cure, and 2 cases of reinfestation after cure), and 5/21 (23.8%) cures, and no cases of reinfestation, in the permethrin group. This difference was not significant ($p = 0.355$) (OR 0.48, 95% CI [0.099–2.319]). The relative severity of failure in most participants was such that four participants were withdrawn from the cocamide DEA group and two from the permethrin group in order to minimise the continuing irritation from infestation. In addition, one person in the permethrin group was lost to follow up without being assessed for efficacy.

In the second study, the single overnight application achieved 10/56 (17.9%) successful treatments (8 cured, and 2 reinfested) compared with 19/56 (33.9%) success (16 cured and 2 reinfested) for the 2 h application regimen repeated after 1 week. This difference was also not significant ($p = 0.0523$) (OR 0.423, 95% CI [0.176–1.020]). One person withdrew from the single treatment regimen immediately after treatment because it was uncomfortable and in the double treatment group there was a drop out and a case of non-compliance in which non-study treatments were used.

Since the outcomes of the second study showed a poor level of efficacy for the cocamide DEA lotion, even with longer application times that *in vitro* tests had indicated would be

**Table 5 Disposition of demographic characteristics of participants in the first clinical study.**

| Characteristic | | Cocamide DEA | Permethrin | Total |
|---|---|---|---|---|
| Number of participants | | 23 | 21 | 44 |
| Age | 4–9 | 12 | 11 | 13 |
| | 10–14 | 5 | 7 | 12 |
| | >18 | 6 | 3 | 9 |
| Median age | | 9 | 9 | 9 |
| Sex | Female | 20 | 15 | 35 |
| | Male | 3 | 6 | 9 |
| Hair features | | | | |
| Length | Short | 3 | 5 | 8 |
| | To shoulder | 9 | 8 | 17 |
| | Below shoulder | 11 | 8 | 19 |
| Thickness | Fine | 7 | 3 | 10 |
| | Average | 9 | 7 | 16 |
| | Thick | 7 | 11 | 18 |
| Curl | Straight | 20 | 13 | 33 |
| | Wavy or curly | 3 | 8 | 11 |
| Dryness | Dry or greasy | 4 | 4 | 6 |
| | Normal | 19 | 17 | 36 |
| Infestation level assessed at enrollment | | | | |
| | Light | 12 | 11 | 23 |
| | Medium | 8 | 7 | 15 |
| | Heavy | 2 | 2 | 4 |
| | Not recorded | 1 | 1 | 2 |

more effective, we decided to run a comparison *in vitro* of the ovicidal effect of the two batches of product. In this series of tests we found that the retained samples of the earlier batch of product (Batch number LI 35101) were more effective than the batch used in the second study (Batch number NH 35101) (Table 7). The differences were highly significant ($p < 0.00001$) for both a 2 h treatment and an overnight treatment, suggesting that if we had used the earlier material in the second trial it could have proved more effective.

Most users of the cocamide DEA 10% lotion reported seeing large numbers of dead lice on their pillow the morning after treatment, and some reported seeing darkly coloured dead lice washed from the hair as the product was rinsed out. This was particularly reported by parents who chose to wash their children's hair in the bath, with the result that the dead insects were seen floating on the surface of the bath water.

In the first study, this was the first experience of head louse infestation for two, and in the second study for five, of the participants. All other participants claimed to have been treated unsuccessfully prior to entry into the study using one or more insecticide or essential oil based products, in most cases alternating with some kind of combing process, including six people from the first study and 27 from the second who previously had used only wet combing with conditioner. Apart from wet combing with conditioner, the

**Table 6 Disposition of demographic characteristics of participants in the second clinical study.**

| Characteristic | | Cocamide DEA 1 × 8 h | Cocamide DEA 2 × 2 h | Total | *p* value |
|---|---|---|---|---|---|
| Number of participants | | 56 | 56 | 112 | |
| Age | 4–9 | 32 | 36 | 68 | NS |
| | 10–14 | 17 | 7 | 24 | 0.038 |
| | >18 | 7 | 13 | 20 | NS |
| Median age | | 9 | 8 | 9 | |
| Sex | Female | 47 | 43 | 90 | NS |
| | Male | 9 | 13 | 22 | NS |
| Hair features | | | | | |
| Length | Short | 11 | 13 | 24 | NS |
| | To shoulder | 17 | 14 | 31 | NS |
| | Below shoulder | 28 | 29 | 57 | NS |
| Thickness | Fine | 15 | 24 | 39 | NS |
| | Average | 24 | 14 | 38 | 0.073 |
| | Thick | 17 | 18 | 35 | NS |
| Curl | Straight | 44 | 40 | 84 | NS |
| | Wavy or curly | 12 | 16 | 28 | NS |
| Dryness | Dry or greasy | 15 | 14 | 29 | NS |
| | Normal | 41 | 42 | 83 | NS |
| Infestation level at enrollment | | | | | |
| | Light | 12 | 11 | 23 | |
| | Medium | 8 | 7 | 15 | |
| | Heavy | 2 | 2 | 4 | |
| | Not recorded | 1 | 1 | 2 | |

**Table 7 Comparison of the two product batches used in the clinical studies for ovicidal activity *in vitro*.**

| Treatment | Time | Number of eggs | | | | Mortality % (undeveloped %) |
|---|---|---|---|---|---|---|
| | | Total | Hatched | Died | Undeveloped | |
| NH 35101 | 2 h | 411 | 357 | 24 | 30 | 13.1% (7.3%) |
| NH 35101 | Overnight | 291 | 100 | 37 | 154 | 65.6% (52.9%) |
| LI 35101 | 2 h | 391 | 39 | 318 | 34 | 90.0% (8.7%) |
| LI 35101 | Overnight | 389 | 0 | 3 | 386 | 100% (99.2%) |
| Control | | 591 | 533 | 16 | 42 | 9.8% (7.1%) |

majority of treatments used also had a conditioner-type base (cetyl alcohol emulsion) so the previous treatments had all left residues of conditioning lipids on the hair, which were completely removed by the cocamide DEA. As a result, all participants/parents except one reported that the treated hair recovered its normal lustre and texture, which had been

Table 8 **Adverse events possibly or probably related to treatment in the two clinical studies.**

| Participant | Treatment | Adverse event | Duration of event |
|---|---|---|---|
| **Study 1** | | | |
| 002 | Permethrin 10 min | Itch when washed off | 5 min |
| 009 | C-DEA 60 min | Itch when washed off | 5 min |
| 011 | C-DEA 60 min | Itch when washed off | 5 min |
| 018 | C-DEA 60 min | Itch during and after treatment | 60 min |
| 019 | C-DEA 60 min | Itch during and after treatment | 60 min |
| **Study 2** | | | |
| 004 | C-DEA 1x O/N | Itch after treatment | 2–3 days |
| 007 | C-DEA 1× O/N | Itch from before treatment | 4 days |
| 008 | C-DEA 1× O/N | Stinging eyes | During treatment |
| 033 | C-DEA 1× O/N | Rash on neck | 4–5 days |
| 070 | C-DEA 1× O/N | Wheals/rash on chest 24 h after treatment | 2–3 days |
| 097 | C-DEA 1× O/N | Intermittent itch | 2–3 days |
| 100 | C-DEA 1× O/N | Intermittent itch | 2–3 days |
| 014 | C-DEA 2× 2h | Itch after treatment | 3 days |
| 034 | C-DEA 2× 2h | Itch after treatment | 1–2 days |
| 051 | C-DEA 2× 2h | Itch during shampooing off | 3–4 min |
| 057 | C-DEA 2× 2h | Itch during shampooing off | 3–4 min |

obscured by the conditioning chemicals, because the product had stripped away all the excess lipid from the hair shafts.

In the second study, bottles of product were weighed before and after use to determine how product much was used during each treatment. This ranged from 23.5 g up to 260.3 g (mean 94.0 g) during a single application, depending upon the length and thickness of the hair.

The raw data for both studies as extracted from the case records are appended as Datas S1 and S2.

### Adverse events

Across the two studies there were 20 reported adverse events. None of these were serious and; apart from two that were attributed to viral infections (1 head cold, 1 sore throat with fever), one of continued itch due to lice from before treatment, and one in which a child was sprayed in the face with an unrelated detergent product by a sibling; all events were some form of application site reaction to the treatment. All these reactions were of short duration and all resolved spontaneously (Table 8).

## DISCUSSION

Cocamide diethanolamine (cocamide DEA) is a powerful surfactant and foaming agent formerly quite widely used in toiletry shampoos. Currently its use in this context is mostly limited to so-called clarifying or stripping shampoos designed to remove lipid and other chemical deposits from hair so that further treatments for permanent waves or colouring can be applied and until recently was considered safe in leave on preparations

containing 10% of the active substance (*Cosmetic Ingredient Review Expert Panel, 1996*). However, since this study was conducted the compound has been reclassified by the *International Agency for Research on Cancer* (a World Health Organization body) in Group 2B, compounds possibly carcinogenic to humans (*International Agency for Research on Cancer, 2013*).

It can be assumed that any activity of cocamide DEA against head lice originates in the surfactant activity of the compound to emulsify some of the lipid protective waterproofing layer of the cuticle of the insects. The result would be that, as the material is rinsed off, the emulsified lipids would be taken with it, leaving the insects susceptible to dehydration because the protective layer on the surface of the cuticle is damaged. Observations of treated lice confirmed that conclusion, initially showing a knockdown-like reaction in which the limbs contracted, and then appearing shrunken and dehydrated when washed from the hair.

Our studies showed that cocamide DEA 10% was able to kill many lice but was ineffective against a proportion of the insects. With the shorter application time of 60 min (study 1) there would have been little concentration effect due to evaporation and with the relatively runny lotion it was possible that not all lice were adequately coated with the fluid and could have survived as a result. However, in the second study both treatment regimens involved deliberate concentration of the lotion on the hair using a hand held dryer, which ensured that all parts of the hair and scalp were coated with the viscous residue. Nevertheless some lice and louse eggs survived the exposure on most participants.

The low level of activity in our studies contrasts starkly with the claims of 100% efficacy with a single 60 min application made in the report of a previous uncontrolled study (*Mülhofer, 1994*). In that study the participants were vigorously brushed, with a bristle brush, so that lice undergoing a knockdown effect were dislodged from the scalp and removed by the investigators. However, when the participants were followed up it is puzzling that no louse eggs were reported to have hatched subsequent to treatment, especially since each of the participants was treated using only 5–10 ml of the product. In contrast, we used over 90 ml for an average treatment in our second study and the treatments were applied for at least double the time of 1 h used in the earlier trial (*Mülhofer, 1994*), which suggests either that cocamide DEA had become significantly less effective against European lice in fewer than 5 years, and in an area where it had not previously been used, or else there was something seriously wrong with the assessment and follow up methodology used in the Austrian study.

The stark contrast in effect observed in our post-trial comparison of the two batches of lotion using *in vitro* testing suggests that there was considerable variability in the active materials used. Both batches were investigational preparations made up in the laboratory of the sponsor, using different batches of raw material. Why such a difference was observed in tests against the same group of louse eggs could not be explained other than as a result of differences in the bioavailability, and activity, of the active material in the two batches of formulation.

The use of surfactants for control of louse infestations is not novel, with folk-lore remedies dating back centuries using materials like saponins as part of treatment. But in the modern era no treatment had employed detergents or surfactants alone for killing the insects until quite recently. However, there are a number of reports of effectiveness of surfactant based materials from the patent literature such as non-vicinal diols (*Lover et al., 1981c*);  aliphatic or aryl aliphatic alcohols and aliphatic esters (*Stafford Miller Ltd, 1979*);  substituted monohydric alcohols (*Lover et al., 1981d*); higher alcohols (*Stafford Miller Ltd, 1982*);  glycine derivatives (*Lover et al., 1981b*); polyoxethylene derivatives (*Lover et al., 1981a*); imidazoline (Miranol$^{TM}$) surfactant compounds (*Lover, Singer & Lynch, 1980*); aminoproprionic acid derivative surfactants (*Lover et al., 1978*), although none of these materials was ever used commercially. Since then one shampoo based on cocamide DEA and other coconut derived surfactants has been reported to show efficacy (*Connolly et al., 2009*). More recently there have been investigations of vicinal diols, a group of compounds little used in any application before the 1990s and which are also derived from coconut, some of which were found to be effective against lice (*Campbell & Carver, 2002*). One of these, 1,2-octanediol, has also been clinically tested and shown to be effective when incorporated into the right vehicle using a 5% concentration (*Burgess et al., 2012*). All of these are believed to act on the surface lipids of the louse cuticle to disrupt the waterproofing layer so that the insects lose water uncontrollably and dehydrate (*Burgess et al., 2012*). From this it can be concluded that surfactants can be effective to kill and eliminate head lice. However, simply making a solution of a powerful surfactant like cocamide DEA, even when subsequently concentrated by evaporation of the water from the mixture, does not necessarily provide adequate activity to disrupt the cuticle lipid on all lice and demonstrates the importance of proper formulation as a major factor in achieving sufficient activity.

## ACKNOWLEDGEMENTS

We would like to thank Dr Angela Owen-Smith, former consultant paediatrician, who acted as local medical contact for the first study, and Dr Nick Irish, former consultant in communicable disease control, who was medical contact for the second study. Peter N Lee provided statistical advice during the design stages of the first study. Investigation team members who contributed to the studies, but who are not named as authors, were Nazma A Burgess, Rachel Cooper, Ann Scarlett, and David Thomson.

### Funding

This study was funded by Riemann a/s, Hillerød, Denmark. The funders had no role in study design, data collection and analysis, decision to publish, or preparation of the manuscript.

## Grant Disclosures

The following grant information was disclosed by the authors:
Riemann a/s, Hillerød, Denmark.

## Competing Interests

The authors declare there are no competing interests.

## Author Contributions

- Ian F. Burgess conceived and designed the experiments, performed the experiments, analyzed the data, contributed reagents/materials/analysis tools, wrote the paper, prepared figures and/or tables.
- Elizabeth R. Brunton performed the experiments, contributed reagents/materials/ analysis tools, reviewed drafts of the paper.
- Christine M. Brown conceived and designed the experiments, performed the experiments, contributed reagents/materials/analysis tools, reviewed drafts of the paper.

## Human Ethics

The following information was supplied relating to ethical approvals (i.e., approving body and any reference numbers):

Huntingdon Local Research Ethics Committee of Cambridgeshire Health Authority. REC reference 97/229, dated 25th July 1997.

Cambridge Local Research Ethics Committee. REC reference 99/211, dated 17th September 1999.

## Patent Disclosures

The following patent dependencies were disclosed by the authors:

Campbell J, Carver A. 2002. Pesticides based on vicinal diols. International Patent; WO 02/069707 Al.

Lover MJ, Singer AM, Lynch DM. 1980. Ectoparasite toxicants containing imidazoline. US Patent, 4,238,499.

Lover MJ, Singer AM, Lynch DM, Rhodes WE 3rd. 1978. Aminoproprionic-acids as ectoparasiticides. US Patent, 4,126,700.

Lover MJ, Singer AM, Lynch DM, Rhodes WE 3rd. 1981a. Use of polyoxethylene derivatives as ectoparasiticides. UK Patent, GB1 604 622.

Lover MJ, Singer AM, Lynch DM, Rhodes WE 3rd. 1981b. Use of glycine derivatives as ovicides or insecticides. UK Patent, GB1 604 854.

Lover MJ, Singer AM, Rhodes WE 3rd, Bilodeau WN. 1981c. Use of certain polyol toxicants as ectoparasiticides or ovicides. UK Patent, GB1 604 856.

Lover MJ, Singer AM, Lynch DM, Rhodes WE 3rd, Bilodeau WN. 1981d. Ectoparasiticidal toxicants. UK Patent, GB1 604 859.

Stafford Miller Ltd. 1979. Pediculicidal toxicants. UK Patent, GB1 547 020.

Stafford Miller Ltd. 1982. Use of higher alcohols as toxicants against lice. UK Patent, GB1 604 857.

## Clinical Trial Registration

The following information was supplied regarding Clinical Trial registration:
NCT02500524 and NCT02499549.

## Supplemental Information

Supplemental information for this article can be found online at http://dx.doi.org/
10.7717/peerj.1368#supplemental-information.

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
