# Peer review of "Laboratory and clinical trials of cocamide diethanolamine lotion against head lice"

_PeerJ, doi:10.7717/peerj.1368_

## Round 0.1 · original submission · Major Revisions

· Academic Editor

Major Revisions

Both reviewers have much appreciated your work. They underlined, however, some issues; in particular, concerning the design of the pre-clinical studies.

Reviewer 1 ·

Basic reporting

1) My native language is not English so my ability to evaluate the proper use of language is limited.
2) The introduction of the article is correct.
3) The structure of the article Submitted conforms to one of the templates.
4) Figures presented by the authors are relevant, but I think there figures to be added if all the studies presented in this first version remain in the article (see below).
5) The submission is 'self-contained', represents an appropriate 'unit of publication', but do not include all results relevant to the hypothesis (see below)

Experimental design

I think the article is struggling in this category.
The article describes an research within the scope of the journal.
But:
1) The article does not clearly define the questions, the methods was not described with sufficient information, and the investigation was not conducted rigorously especially when considering the in vitro/pre-clinical studies for which it is not clear their inclusion in the article. While the introduction states that "At each stage the materials was tested in vitro to establish the approach to dosing followed by two clinical investigations using different treatment regimens," is not clear how the results of these studies contributed to the design of subsequent clinical trials, especially when the first of the pre-clinical studies evaluated only one dose.
2) In addition, these pre-clinical studies have design problems. The first one (lines 91-96) did not have (or at least is not described) control group and only assessed one exposure time; thus the bioassay had one treatment which has no comparison with any other. Given this (no controls and no other treatment), the purpose of this assay is unclear. In the second pre-clinical study (lines 98-102) it is not clearly established how the following variables were evaluated: the influence of hair drying compared with humidification after application of the product, and the sensitivity of louse eggs of different ages to the product. Nor it is established if were evaluated post-embryonic stages (nymphs/adults) and eggs or only eggs (the results section does not help to clarify this). Again, there is no reference to the realization of controls.
3) The clinical studies are optimally described (104-229). Overall the manuscript has greater care in describing these studies compared to more experimental studies (pre-clinical or in vitro studies), so perhaps the manuscript can do without in vitro studies and only focus on the description and discussion of the clinical studies.
4) The authors state that the sample sizes were calculated by considering specific statistical parameters (power, confidence, etc) (lines 203-210), however those sample sizes were not achieved (as described by the authors, lines 152-153 and 161-162), but the authors do not say anything about the final performance of the used statistical tests.

Validity of the findings

Again, I think the article has difficulty in this category
1) The results of pre-clinical studies, besides providing little information to work, are not properly presented (lines 233-250). There are shortages (almost no) of numerical data, quantitative comparisons, statistical differences, or tables. Clearly is not conceivable an experimental publication without numerical data presented in tables or in the text.
2) As is clear from what I wrote above, the results of the first pre-clinical study (the only where percentage of effect are presented) (lines 233-237) has no contribution since there is no control or treatment to compare.
3) Lines 239-240 should go in the discussion section
4) Lines 242-244 and 246-250. It is not clear the origin of these results, there is no reference to any table and are not clearly described in the text. I interpret corresponding to the second in vitro study, but this must be well clarified in the text, together with the concrete description of the results and a table.
5) In summary, the authors should re-write the presentation of the results of pre-clinical studies. Moreover, and more important, the authors should state clearly the contribution of these studies to the work as a whole and to the subsequent clinical studies particularly, or the authors should reconsider the report of the in vitro studies.
6) Lines 282-289. The authors provide the results of a new laboratory study, explaining the need to do it, but do not provide a description of the methodology, either in this section or M and M. This study should be described in M and M. Paradoxically, the authors present a table of the results of this study but do not present tables with the results of the in vitro studies described in M and M.
7) As for the description of the experimental design, the results of clinical studies are presented more correctly compared with the results of in vitro studies.
8) The discussion is focused on the clinical trials and is the most accomplished of the manuscript which reinforces my comment and advice to eliminate pre-clinical studies of the work.

·

Basic reporting

This article basically described two clinical tests to evaluate the effectiveness of a surfactant-based lotion containing 10% cocamide diethanolamine (cocamide DEA) against Pediculus humanus capitis (head lice). In the first test, the activity of the lotion (1 h exposure) was compared with the activity of 1% permethrin crème rinse. In the second test, the exposition time was increased (overnight) or the number of treatments was increased (2 applications in 7 days).

The objective of the article is relevant because of the worldwide importance of pediculosis capitis, and the ineffectiveness of numerous commercial products due to unsuitable formulations and the evolution of insecticide resistance in head louse populations.

The results showed low effectiveness of the 10% cocamide DEA lotion to reduce the head louse infestation, either with 1 hour exposure or when it was concentrated by hair drying. These results strongly contrast with the claims of 100% efficacy previously reported by other authors.

The title is adequate for the study, the introduction state the problem being investigated and methodology and results are summarized in four tables.

Experimental design

The methodology was a randomized, controlled, assessor blinded clinical study. This methodology has been previously used by the authors, who have extensive experience in clinical studies of pediculicide products.

The immersion methodology (10 seconds) used to assess the in vitro activity of the 10% cocamide DEA lotion on Pediculus humanus humanus (body lice), is appropriate to the objective of the trial.

Validity of the findings

No comments

Comments for the author

Some corrections should be done and some information should be done to improve the manuscript.

Preclinical studies: they were performed by immersion of groups of the body lice in the tested products. According to Line 92 (methods), the insects or eggs were exposed for 10 seconds to the products, but according to Line 23 (abstract) and Line 233 (results) the application time was 60 minutes. Please, clarify.
Also, more information about the insects should be added, such as the number of body lice used in each assay and the development conditions (nymphs?, adults?, a mix ?, starved?).

Second study: Authors said that after the application of the product, they evaporated the excess of water using a hair drying (Line 123). Please, add what was the estimated time it took to dry hair.

Participants: the possible participants were checked for living lice using a plastic detection comb (line 139). Have authors made some selection based on the level of infestation? Have you included individuals with low (1- 2 lice) or high (20-100 lice) infestation? Probably, the level of infestation affects the efficacy of the treatment.

Table 1 and Table 2 showed demographic characteristics in the first and second clinical study. Some of the hair features are easy to define (length, curl), but others are difficult to classify (thickness, dryness). Authors should add some general guidelines they used to do this hair classification.

Table 3: the highly significant differences in the ovicidal effect of the two batches of the cocamide DEA lotion (table 3) are surprising and certainly they affected the results of the trials. Have authors any explanation or justification for such differences? Did you use commercial products?. Please comment some possible reasons for the different ovicidal activities measured in the different batches.

---

## Round 0.2 · Minor Revisions

· Academic Editor

Minor Revisions

Dear authors, both reviewers have much appreciated your efforts in improving the manuscript. One of the reviewers suggests that you better explain the tests on eggs. Even though this information is included in material and methods, the way it is provided is not completely clear for the reader.

Reviewer 1 ·

Basic reporting

The manuscript has been successfully modified according to my previous comments. The description of the methodology and results of pre-clinical assays (main objective of the prior reviews) have been significantly improved as a result of the changes and additions made by the authors.
However, in order for the reader to understand without difficulty and clearly the experiments, I make minimum final comments on the presentation of these assays. If the Editor understands that the changes, once made by the authors, meet the improvements requested by me it is not necessary that the manuscript is sent to me again. The comments are at the Experimental Desgin section of this review.

Experimental design

1) Respect to second pre-clinical study: as it is apparent from the description of the results (lines 313-335) and tables 3 and 4, these studies were conducted only on eggs. However, materials and methods (lines 122-136) is not clear that this study was conducted only on eggs. Please clarify this point.
2) In Tables 1, 2, 3 and 4 are presented categories to characterize the toxic effect (killed, mobile, immobile, hatched, half-hatched, died, undeveloped), but these were not described in Materials and Methods. The authors need to describe these categories, as well as the criteria for determining them, and make it clear which of them were considered to determine mortality (e.g. I understand that egg mortality = died + undeveloped)

Validity of the findings

No comments

·

Basic reporting

This article described laboratory and clinical tests to evaluate the effectiveness of a surfactant-based lotion containing 10% cocamide diethanolamine (cocamide DEA) against Pediculus humanus capitis (head lice)
The objective of the article is relevant because of the worldwide importance of pediculosis capitis, and the ineffectiveness of numerous commercial products due to the evolution of insecticide resistance.
The results show low effectiveness of the 10% cocamide DEA lotion to reduce the head louse infestation, either 1 hour exposure or when it was concentrated by hair drying. These results strongly contrast with the claims of 100% efficacy previously reported by other authors.
The title is adequate for the study, the introduction state the problem being investigated and methodology and results are summarized in four tables.

Experimental design

The methodology basically is a novel use of a technique applied to extract an insecticide from an insect fluid. This solvent-less methodology to extract deltamethrin from each lipoprotein fraction showed high sensitivity performance and could be also used in small insects. The research must have been conducted in conformity with the prevailing ethical standards in the field.

Validity of the findings

No comments

Comments for the author

All suggested corrections have been done and all missed information had been added in this new version of the manuscript.
I consider that the article should be accepted for publication in PeerJ journal.

---

## Round 0.3 · accepted · Accept

· Academic Editor

Accept

Thank you very much for the corrections. The experimental procedure is now clearer and more informative.